# VF-Mamba: See What Matters First for Remote Sensing Semantic Segmentation

## Abstract

While Mamba models offer efficient global modeling, fixed scanning limits adaptation in complex scenes. This limitation is critical for remote sensing segmentation, where fine-grained understanding of rare objects is required. In this paper, we propose Visual Focus Mamba (VF-Mamba), a saliency-guided state space model that scans informative regions first. Our VF-Mamba is inspired by the phenomenon that the human visual system prioritizes salient regions for progressive scene interpretation. Based on this principle, we develop Saliency-Driven Scanning (SDS), which ranks patches according to their distributional distinctiveness. The underlying foundation of our approach is that samples exhibiting substantial distributional disparities are to be classified as either sparse categories or regions of interest, thus prioritizing scanning them can improve segmentation performance. For the purposes of enhancing robustness and efficiency, respectively, two strategies are hereby proposed: Full-SDS employs pairwise Wasserstein distances for precise measurement, while Sparse-SDS utilises a Gaussian reference for efficient approximation. The efficacy of VF-Mamba is evaluated using two benchmark datasets. The results demonstrate higher accuracy and better preservation of fine details in small classes compared to state-of-the-art methods, achieved a 3.28% IoU improvement on challenging categories.

## 1 Introduction

Semantic segmentation (Xu et al., 2024; Dong et al., 2023) is a fundamental task in computer vision, where each pixel in an image is assigned as label, enabling pixel-level classification that supports further scene analysis. The objective of remote sensing semantic segmentation (Pan et al., 2025; Ye et al., 2025) lies in facilitating the acquisition of insights into natural environments and human activities. This technology finds application in a variety of contexts, including environmental monitoring (Li et al., 2022), land resource management (Yan et al., 2022), urban planning (Zhou et al., 2023), and disaster response (Yang et al., 2024).

In recent years, numerous semantic segmentation models have been developed for remote sensing images, which feature large sizes, complex backgrounds, diverse categories, and long-tail distributions. CNN-based methods like UNet (Ronneberger et al., 2015), DeepLabV3+ (Chen et al., 2018), PSPNet (Zhao et al., 2017), and FPN (Kirillov et al., 2019) employ encoder-decoder structures with upsampling (Fu et al., 2025) to restore feature maps but remain limited in modeling global semantics and long-range dependencies due to local receptive fields. The success of Vision Transformers (ViT) (Dosovitskiy et al., 2020) has advanced the field, with architectures such as UNetFormer (Wang et al., 2022) achieving strong results. However, the quadratic complexity of ViT-based methods (Xu et al., 2024; Su et al., 2025) poses significant computational challenges for large-scale remote sensing data. Mamba (Gu & Dao, 2023), a recent state space model, offers linear complexity while retaining global context modeling capabilities. Mamba-based models (Liu et al., 2024; He et al., 2025; Li et al., 2024) have shown promising results in natural image segmentation, and RS3Mamba (Ma et al., 2024) demonstrated its potential in remote sensing. Yet, existing methods employ fixed-direction, uniform scanning (Liu et al., 2024; Li et al., 2025; Pei et al., 2025), restricting the construction of content-aware, spatially adaptive representations. This limits progressive visual understanding, which is vital for remote sensing semantic segmentation that demands integrating local details into global semantics.

Figure 1: (a) Ground truth labels. (b) Sequential scanning. (c) Saliency-driven scanning. Red marks critical, rare-category regions; green indicates moderately important areas; blue denotes peripheral regions. The numbers indicate the scanning order. Saliency-driven scanning prioritizes key regions, enriching semantics and improving segmentation in complex scenes.

In this paper, we propose Visual Focus Mamba (VF-Mamba), a novel visual-adaptive state space model that incorporates saliency-driven scan (SDS) into the Mamba framework for semantic segmentation of remote sensing images, inspired by the human visual perception system. Humans instinctively focus on salient regions characterized by distinct contrast, texture, or semantic cues. This selective attention enables hierarchical scene interpretation, progressively incorporating context to form high-level understanding. In Mamba, the scanning order directly influences the semantic quality of internal representations. Prioritizing informative regions early in the sequence provides stronger semantic context, improving the interpretation of subsequent areas, as illustrated in Figure 1. VF-Mamba guides the scanning process based on content saliency, prioritizing visually distinctive regions and progressively building semantic representations, thereby aligning with human visual perception and improving remote sensing semantic segmentation accuracy and efficiency. Specifically, we introduce two SDS strategies, **Full-SDS** and **Sparse-SDS**, which transform patch-wise feature distributions into saliency scores for adaptive ranking. In Full-SDS, we compute the Wasserstein distance (WD) between the embedding of each patch and those of all other patches, capturing precise distributional differences to serve as saliency scores. Sparse-SDS first computes the mean and variance of the feature map and generates random Gaussian noise, then measures the WD between each patch embedding and this noise to efficiently estimate distributional differences.Our method focuses solely on the statistical distributions of patches, enabling robust estimation of saliency scores that effectively guide Mamba's progressive understanding of both scene structures and semantic information within images.

In summary, our contributions are as follows:

- We propose VF-Mamba, a saliency-driven model for remote sensing semantic segmentation that replaces fixed scanning order with a flexible, content-adaptive strategy inspired by the human visual system, enabling progressive visual understanding.
- We introduce Saliency-Driven Scanning (SDS) to connect one-dimensional sequential processing with two-dimensional spatial analysis. We design two novel strategies, Full-SDS and Sparse-SDS. They convert patch-level distribution differences into saliency scores for adaptive scanning. This represents a new approach for precise and efficient feature prioritization.
- We validate VF-Mamba through extensive experiments on the ISPRS Potsdam dataset and the ISPRS Vaihingen dataset, our method produces optimal results on both datasets. Comprehensive ablation studies further demonstrate the effectiveness of our approach.

## 2 RELATED WORK

### 2.1 CONVOLUTIONAL NEURAL NETWORK

CNNs have been fundamental in vision tasks like classification, detection, and restoration since AlexNet (Krizhevsky et al., 2012). Enhancements such as depth-wise (Chollet, 2017; Howard et al., 2017), deformable convolutions (Dai et al., 2017), residual connections (He et al., 2016), and attention (Hu et al., 2018; Jaderberg et al., 2015) have improved efficiency and representation. In

segmentation, models like UNet (Ronneberger et al., 2015) and DeepLab (Chen et al., 2014; 2018) capture hierarchical features well. However, their limited receptive fields hinder long-range context modeling, which is crucial for understanding complex remote sensing scenes.

## 2.2 VISION TRANSFORMERS

Transformer (Vaswani et al., 2017) has transformed vision tasks by modeling long-range dependencies through self-attention. ViT (Dosovitskiy et al., 2020) achieves strong classification performance but relies on large-scale pre-training. DeiT (Touvron et al., 2021) mitigates this with CNN-based distillation. For segmentation, SETR (Zheng et al., 2021), SegFormer (Xie et al., 2021), and Swin Transformer (Liu et al., 2021) introduce hierarchical structures and window attention, boosting performance on both natural and remote sensing datasets. To reduce self-attention cost, linear attention (Katharopoulos et al., 2020), shifted windows, and sparse attention (Child et al., 2019; Yuan et al., 2025) have been proposed. Yet, ViT-based models struggle to capture spatially adaptive context under limited resources, which is crucial for fine-grained remote sensing segmentation.

## 2.3 STATE SPACE MODEL FOR VISION

State Space Models (SSMs) provide efficient alternatives to Transformers by modeling long-range dependencies with linear complexity. Early models like S4 (Gu et al., 2021) used structured transitions with diagonalized parameters. Mamba further improved this by introducing selective state updates for linear-time global modeling. To apply Mamba to vision tasks, models such as VMamba (Liu et al., 2024), FEAST-Mamba (Li et al., 2025), and EfficientVMamba (Pei et al., 2025) adopt multi-directional scanning for 2D data. DefMamba (Liu et al., 2025) used a content-aware deformable scanning strategy. These methods perform well on natural image segmentation but still struggle with fine-grained details, scale variation, and adaptive context modeling in complex remote sensing scenes.

## 2.4 MAMBA FOR REMOTE SENSING

Mamba has driven progress in remote sensing semantic segmentation. RS3Mamba (Ma et al., 2024) first applied Mamba to this field, utilizing Visual State Space (VSS) for efficient global modeling and outperforming CNNs and ViTs. Samba (Zhu et al., 2024b) enhances VSS with MLPs for better representation, while UNetMamba (Zhu et al., 2024a) uses VSS in the decoder alongside CNNs to improve local feature perception. UrbanSSF (Wang et al., 2025) combines Transformer-based feature extraction with Mamba for multi-scale fusion. Despite their success, these methods rely on pre-defined VSS modules with fixed scanning orders, limiting adaptability to image content. This constraint reduces flexibility in complex scenes and highlights the need for more adaptive scanning strategies.

# 3 METHOD

## 3.1 STATE SPACE MODEL

SSMs inspired by control systems, can be viewed as Linear Time-Invariant (LTI) systems. Given the system input $x(t) \in \mathbb{R}^L$ where $L$ is the sequence length, the system output is expressed as

$$h'(t) = \mathbf{A}h(t) + \mathbf{B}x(t), \tag{1}$$

$$y(t) \in \mathbb{R} = \mathbf{C}h(t), \tag{2}$$

where $h(t) \in \mathbb{R}^M$ represents the hidden state, $M$ is the number of states, $\mathbf{A} \in \mathbb{R}^{M \times M}$, $\mathbf{B} \in \mathbb{R}^{M \times 1}$, and $\mathbf{C} \in \mathbb{R}^{1 \times M}$ are the weighting parameters.

Subsequently, to integrate the SSM into neural networks, it is necessary to discretize the continuous-time SSM. Specifically, Mamba employs a zero-order hold mechanism and introduces a time scale $\Delta$ to convert the continuous matrices $\mathbf{A}$ and $\mathbf{B}$ into their discrete counterparts $\overline{\mathbf{A}}$ and $\overline{\mathbf{B}}$ by

$$\overline{\mathbf{A}} = \exp^{(\Delta A)}, \tag{3}$$

$$\overline{\mathbf{B}} = (\Delta \mathbf{A})^{-1}(\exp^{(\Delta \mathbf{A})} - \mathbf{I}) \cdot \Delta \mathbf{B}, \tag{4}$$

where $\mathbf{I} \in \mathbb{R}^{M \times M}$ denotes the identity matrix. Then, the discrete formulation of the SSM, obtained through zero-order hold discretization, is expressed as

$$h_t = \overline{\mathbf{A}} h_{t-1} + \overline{\mathbf{B}} x_t, \tag{5}$$

$$y_t = \overline{\mathbf{C}} h_t, \tag{6}$$

where $\overline{\mathbf{A}}$ is initialized as a learnable diagonal matrix, and together with $\overline{\mathbf{B}}$ and $\mathbf{C}$ collectively form a discrete LTI system.

## 3.2 NETWORK ARCHITECTURE

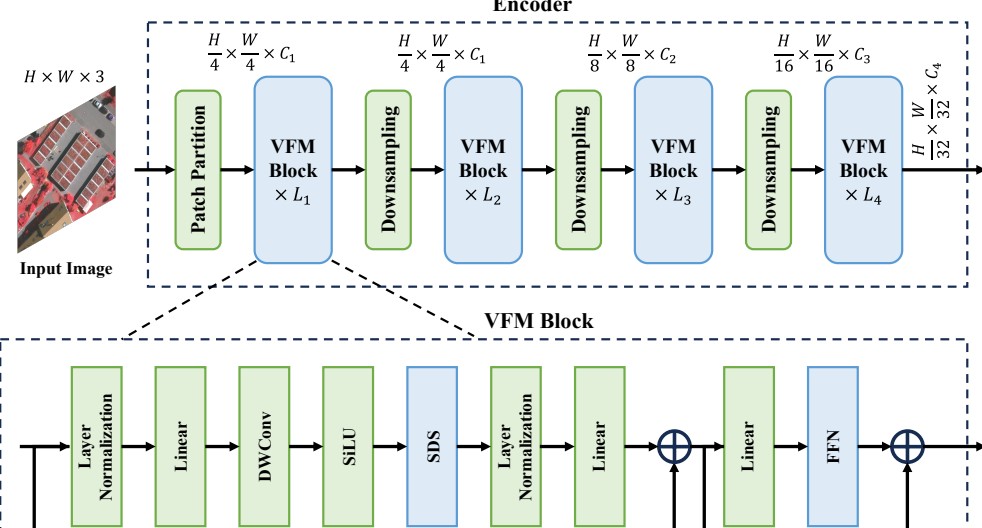

Figure 2: Schematic diagram of the VF-Mamba framework. The encoder consists primarily of Patch Partitioning, VF-Mamba blocks, and downsampling operations. The core computation within each VFM block is performed by the SDS module. For decoding, networks such as UPerNet are employed.

As shown in Figure 2, the encoder of VF-Mamba is built upon VMamba-T (Liu et al., 2024), with the original VSS block replaced by the proposed VF-Mamba (VFM) block. The core of the VFM block is the proposed SDS mechanism, where either Full-SDS or Sparse-SDS is used exclusively. For the decoder, the UPerNet (Xiao et al., 2018) architecture is employed to generate the semantic segmentation results.

## 3.3 VISUAL FOCUS MAMBA BLOCK

As illustrated in Figure 2, after feeding the input image into the network, a patch partition operation is first applied, followed by the addition of positional embeddings. The purpose of positional embeddings is to provide the model with spatial location information, which facilitates the preservation of spatial relationships among patch blocks. The resulting patch features are then passed into the VFM block for further feature extraction. The VFM block serves as a crucial component for feature representation and is constructed by modifying the scanning process. Selective scanning, which lies

Figure 3: Illustration of the saliency-driven scan. Input patches are processed through either Full-SDS or Sparse-SDS to compute a distance matrix, which is then used to rank the patches. The sorted patches are subsequently fed into the S6 block for progressive visual understanding in Mamba.

at the core of Mamba, enables global receptive fields and dynamic weighting with linear computational complexity. In our framework, the newly proposed SDS module is employed as the selective scanning mechanism within the VFM block, allowing the model to capture salient and semantically progressive information. Further details of this module are described in the following subsection.

## 3.4 FULL SALIENCY-DRIVEN SCAN

There exist numerous methods for identifying the most salient and distinctive regions within an image, such as using Euclidean distance, cosine similarity, and other measures. Our approach involves computing the similarity between patches to quantify differences in their distributions, and subsequently identifying the patch with the largest distributional discrepancy as the most salient region. The patches can then be ranked based on these quantified distributional differences and sequentially fed into the S6 module for scanning. In remote sensing imagery, regions that are visually salient and distinctive often correspond to rare yet semantically critical classes or objects. Prioritizing the scanning of these regions enables the model to capture features associated with high semantic value at earlier stages, thereby improving segmentation performance in complex scenes.

However, remote sensing images often exhibit complex backgrounds, diverse object categories, and significant variations in the appearance of the same object under different illumination conditions. Additionally, maps in remote sensing images tend to display more pronounced geometric structures. As a result, directly calculating spatial distances using methods like Euclidean distance can be inaccurate, particularly in urban regions affected by shadows. On the other hand, objects belonging to different classes in remote sensing images typically exhibit substantial differences in their probability distributions. Therefore, employing distributional divergence measures provides a more effective means of identifying salient regions.

Considering computational efficiency and interpretability, we adopt the Wasserstein Distance (WD) to measure distributional differences. WD accounts for both the probability discrepancies between the distributions of two patches and the spatial distances between them, while also providing smoother and more stable results on discrete data.

As shown in Figure 3, in the Full-SDS strategy, we comprehensively compute the WD between each patch and all other patches, resulting in a distance matrix. We then calculate the total distance of each patch to all others as a saliency metric, where a larger total distance indicates that the patch is more salient and distinctive within the entire image. Specifically, for the input patch sequence $X = \{x_1, \ldots, x_L\}$, where $x \in \mathbb{R}^{1 \times N}$ has $N$ channels, the WD between patch $x_i$ and patch $x_j$ is computed according to

$$d_{ij} = \frac{1}{N} \|\phi(\text{sort}_L(x_i)) - \phi(\text{sort}_L(x_j))\|_1, \tag{7}$$

$$\phi(x_i) = \{\phi^{(n)}|\phi^{(n)} = \sum_{k=1}^{n} x_{i,k}, n = 1, \ldots, N\} \in \mathbb{R}^{1 \times N}, \tag{8}$$

where $d_{ij}$ denotes the WD between $x_i$ and $x_j$, $\text{sort}_L(\cdot)$ indicates sorting in ascending order, $\|\cdot\|_1$ denotes the $l_1$-norm, and $x_{i,k}$ represents the $k$-th element of $x_i$. By aggregating the values of $d_{ij}$, a distance matrix $D \in \mathbb{R}^{L \times L}$ is constructed. Subsequently, for each patch, the total distance is calculated from matrix $D$ and used as its saliency score according to

$$s_i = \sum_{k=1}^{L} d_{ik}, \tag{9}$$

where $s_i$ denotes the saliency score of $x_i$. The input patches are then ordered based on their corresponding saliency scores by

$$\hat{X} = \text{sort}_H(X, S), \tag{10}$$

where $\text{sort}_H(\cdot)$ denotes the descending order of patches based on saliency scores, and $S$ denotes the collection of these scores. Finally, the saliency-driven sorted patches are fed into the S6 module for processing, and the original order is restored based on the indices, yielding the output of Full-SDS. In summary, Full-SDS leverages precise distributional differences between patches through pairwise Wasserstein Distance computation, enabling accurate identification and prioritization of salient regions. This approach enhances the model's ability to capture critical semantic information, thereby improving segmentation performance in complex remote sensing scenes.

### 3.5 SPARSE SALIENCY-DRIVEN SCAN

Considering that Full-SDS requires pairwise computation of the WD, which leads to high computational complexity when dealing with multiple patches, we additionally propose a sparse variant called Saliency-driven Scan. Since WD measures the difference between probability distributions and is insensitive to the order of samples, its result remains unchanged even if the pixel order in an image is rearranged, as long as the overall distribution of pixel values stays the same. Given that pixel values in remote sensing images generally approximate a Gaussian distribution, we can leverage the mean and variance of the original image to generate a random Gaussian image that shares similar statistical properties. This synthetic image is then used for WD computation to enable distribution-level comparison.

Based on this insight, we propose Sparse-SDS, which generates a random Gaussian image from the mean and variance computed over all patches and computes the WD between this synthetic reference and each original patch, achieving linear computational complexity. Specifically, we first extract the statistical features (mean and variance) from all input patches, according to

$$\mu = \text{mean}(X), \tag{11}$$

$$\theta = \text{std}(X), \tag{12}$$

where $\mu$ denotes the mean, $\theta$ denotes the standard deviation, $\text{mean}(\cdot)$ represents the calculation of the mean, and $\text{std}(\cdot)$ represents the calculation of the standard deviation. Subsequently, a Gaussian image is randomly generated based on these parameters by

$$G \mathcal{N}(\mu, \theta^2) \in \mathbb{R}^{L \times N}. \tag{13}$$

Then, compute the WD between each patch and the corresponding region in $G$, according to

$$d_i = \frac{1}{N} \|\phi(\text{sort}_L(x_i)) - \phi(\text{sort}_L(g_i))\|_1, \tag{14}$$

where $g_i$ denotes the $i$-th patch of $G$. And $d_i$ can be directly used as the saliency score $s_i$ of $x_i$ for sorting. The input patches are then ordered based on their corresponding saliency scores by

$$\hat{X} = \text{sort}_H(X, S). \tag{15}$$

Table 1: Comparison results (IoU, Acc %) on the Vaihingen dataset. Bold indicates the best, red denotes the second-best.

| Method | Class | | | | | | mIoU | mAcc |
| | Imp.surf | Building | Low.veg | Tree | Car | Clutter | | |
|---|---|---|---|---|---|---|---|---|
| DeeplabV3+ | 84.97, 92.75 | **91.71**, **95.83** | **70.70**, 79.97 | 78.65, 92.16 | 72.93, **84.88** | 3.91, 3.93 | 67.48 | 74.92 |
| PSPNet | 84.30, 91.54 | 89.91, 95.26 | 69.21, 79.55 | 78.79, 90.82 | 70.50, 79.50 | 31.94, 36.26 | 70.77 | 78.82 |
| VMamba | 84.54, **93.69** | 88.64, 93.20 | 70.36, 79.77 | **79.29**, 91.04 | 62.92, 77.94 | 26.66, 26.90 | 68.73 | 77.09 |
| DefMamba | 78.67, 91.79 | 80.77, 87.09 | 64.47, 74.55 | 75.83, 90.08 | 48.51, 61.38 | 14.32, 14.65 | 60.43 | 69.92 |
| UNetMamba | 71.29, 77.23 | 68.07, 91.97 | 60.43, 75.39 | 72.98, 81.70 | 34.58, 46.89 | 12.54, 17.57 | 53.32 | 65.13 |
| MSEONet | 82.47, 93.02 | 88.30, 92.92 | 70.36, 79.80 | 78.65, 90.86 | 71.41, 82.02 | 16.35, 16.51 | 67.69 | 75.60 |
| Full-SDS | **85.07**, 92.62 | 89.77, 94.25 | 70.22, **80.38** | 79.17, 91.35 | **73.25**, 84.32 | 32.17, 32.59 | 71.61 | 79.25 |
| Sparse-SDS | 83.92, 91.74 | 88.64, 95.11 | 69.15, 77.12 | 79.26, **92.65** | 73.16, 82.68 | **36.71**, **37.23** | **71.81** | **79.42** |

Finally, the saliency-driven sorted patches are fed into the S6 module for processing, and the original order is restored based on the indices, yielding the output of Sparse-SDS. Compared to Full-SDS, which offers precise but expensive pairwise comparisons, Sparse-SDS provides a scalable solution by approximating global distributional contrast through a synthetic Gaussian reference, striking a balance between efficiency and effectiveness.

## 4 EXPERIMENTS

**Datasets and Metrics** We evaluate our method on two widely used high-resolution benchmarks from the ISPRS 2D Semantic Labeling Challenge: the Potsdam (ISPRS, 2013a) and Vaihingen (ISPRS, 2013b) datasets. To comprehensively evaluate the performance of the proposed model, we adopted the following evaluation metrics: mean accuracy (mAcc) and mean intersection over union (mIoU).

**Implementation Details** The model is trained using pixel-wise binary cross-entropy loss in PyTorch. AdamW (Loshchilov & Hutter, 2017) serves as the optimizer, with learning rates set to $2 \times 10^{-6}$ for fixed components and $2 \times 10^{-4}$ for randomly initialized ones. Training proceeds for 160,000 iterations with a batch size of 4.

**Comparison Methods** We select six representative and state-of-the-art methods spanning convolutional, attention-based, and state space paradigms for performance comparison. These include: 1) Widely-adopted industry standards: DeepLabV3+ (Chen et al., 2018) and PSPNet (Contributors, 2020) , 2) Recent SOTA in state-space models: VMamba (Liu et al., 2024), DefMamba (Liu et al., 2025) and UNetMamba (Zhu et al., 2024a) , and 3) Specialized remote sensing solutions: MSEONet (Huang et al., 2025) , collectively covering the performance frontier in semantic segmentation. All methods conduct comparison experiments under the same implementation details, following their respective optimal parameters settings.

Table 2: Comparison results (IoU, Acc %) on the Potsdam dataset. Bold indicates the best, red denotes the second-best.

| Method | Class | | | | | | mIoU | mAcc |
| | Imp.surf | Building | Low.veg | Tree | Car | Clutter | | |
|---|---|---|---|---|---|---|---|---|
| DeeplabV3+ | **86.63**, **93.22** | 91.82, 96.64 | 74.48, 88.24 | **78.80**, **86.20** | 89.58, 95.23 | 39.75, 47.89 | 76.84 | 84.58 |
| PSPNet | 84.40, 92.30 | 90.99, 96.54 | 74.09, 85.54 | 76.46, 86.08 | 87.46, 94.72 | 38.08, 47.42 | 75.25 | 83.77 |
| VMamba | 82.03, 91.10 | 88.21, 95.02 | 72.33, 85.95 | 69.88, 81.19 | 84.96, 92.57 | 30.87, 36.64 | 71.38 | 80.41 |
| DefMamba | 80.65, 91.49 | 87.24, 93.21 | 68.51, 86.11 | 64.51, 74.38 | 85.49, 93.25 | 24.01, 28.52 | 68.40 | 77.83 |
| UNetMamba | 80.33, 89.09 | 85.43, 91.48 | 68.90, 78.82 | 65.07, 79.31 | 83.61, 88.89 | 28.62, **59.77** | 68.66 | 81.23 |
| MSEONet | 82.44, 92.07 | 87.14, 90.16 | 71.95, **90.20** | 76.24, 85.97 | 89.10, **96.65** | 19.57, 22.60 | 71.07 | 79.61 |
| Full-SDS | 85.95, 92.86 | **91.83**, 96.26 | **76.14**, 89.35 | 77.74, 85.91 | **90.71**, 95.06 | 40.97, 49.18 | **77.22** | 84.77 |
| Sparse-SDS | 85.53, 91.90 | 91.19, **96.78** | 75.93, 88.23 | 77.27, 86.32 | 89.87, 94.23 | **43.03**, 52.60 | 77.14 | **85.01** |

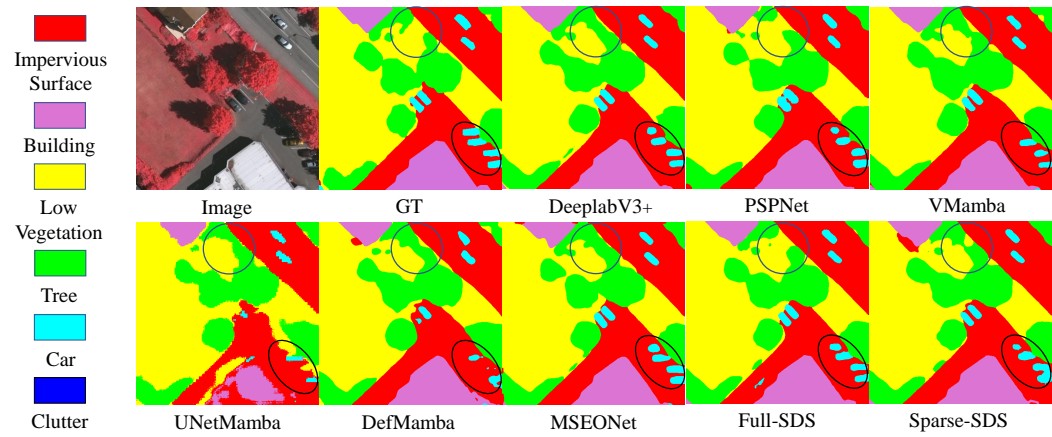

Figure 4: The results on the Vaihingen dataset. VF-Mamba excels in preserving fine-scale structures.

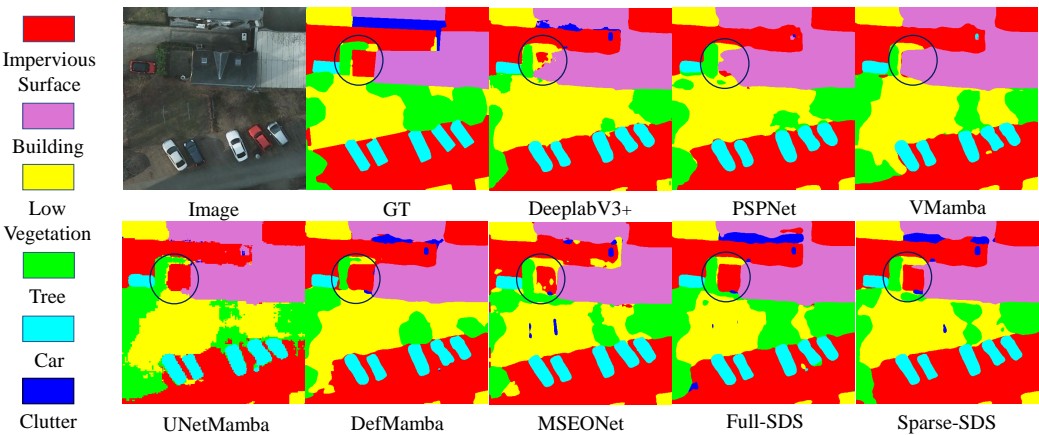

Figure 5: The results on the Potsdam dataset. VF-Mamba excels in preserving geometric regularity and boundary integrity.

### 4.1 QUANTITATIVE EVALUATION

#### 4.1.1 COMPARISON ON THE VAIHINGEN DATASET

Our method demonstrates significant advancements on the Vaihingen benchmark, establishing new state-of-the-art performance across critical metrics. As shown in Table 1, Our Full-SDS achieves a remarkable 71.61% mIoU, surpassing VMamba by 2.88%. Notably, both proposed variants exhibit robust dominance: Full-SDS and Sparse-SDS deliver 71.61% and 71.81% mIoU respectively, consistently exceeding baselines. While the IoU and Acc of other categories remain excellent, vehicle recognition reaches 73.25% IoU. This improvement is attributed to the SDS module, which facilitates earlier focus on rare categories. The 36.71% IoU for challenging Clutter regions highlights exceptional open-set capability. These results confirm that our model achieves balanced precision across diverse terrain features while maintaining superior performance in critical categories.

#### 4.1.2 COMPARISON ON THE POTSDAM DATASET

Consistent superiority is observed on the Potsdam dataset (Table 2), where Full-SDS again leads with 77.22% mIoU, outperforming VMamba by 5.84%. And Sparse-SDS leads with 85.01% mAcc, outperforming VMamba by 4.60%. While the IoU and Acc of other categories remain excellent, clutter recognition reaches 43.03% IoU and vehicle recognition reaches 90.71% IoU, which are also

higher than other algorithms. Overall, the proposed SDS module effectively enhances semantic segmentation performance in remote sensing images, especially for rare yet critical categories. This can be attributed to our statistical analysis of remote sensing data, which reveals that objects like vehicles often exhibit distinctive distribution patterns in the original images, making them more likely to be prioritized by SDS.

### 4.1.3 QUALITATIVE EVALUATION

Figure 4 , 5 present qualitative comparisons on the Vaihingen and Potsdam datasets. On Vaihingen, VF-Mamba accurately segments vehicles (cyan) with clear shapes and boundaries; on Potsdam, it delineates clustered vehicles and narrow roads (red) with high geometric precision in dense urban scenes. This is attributed to the SDS mechanism, which mimics human progressive perception by adaptively focusing on key regions, such as anomalous objects and structural anchors, early in the scanning process. Unlike fixed-order models such as VMamba, VF-Mamba suppresses early-stage errors and maintains spatial coherence across varying scene complexity. Both Full-SDS and Sparse-SDS achieve competitive visual results. Full-SDS offers finer boundary refinement, while Sparse-SDS delivers comparable quality with lower cost, excelling in large-structure integrity and small-object recovery.

### 4.2 ABLATION STUDY

To validate the effectiveness of the proposed SDS module, including Full-SDS and Sparse-SDS, we conduct ablation experiments with four variants: (1) Random, where patch order is randomly shuffled before entering the S6 module, to test the importance of scanning order; (2) Linear, which uses the original sequential patch order, further assessing the benefit of adaptive ordering; (3) Euclidean Distance, replacing WD with Euclidean distance to evaluate the impact of distributional metrics; and (4) Standard Gaussian, which replaces the learned Gaussian distribution in Sparse-SDS with a standard Gaussian to examine the robustness of the approximation strategy.

As shown in Table 3, each module contributes to the strong performance of VF-Mamba. The SDS module enables accurate focus on key regions, improving rare category detection, and alleviating class imbalance. WD models global distributions and ranks patches by distributional differences, allowing Full-SDS to capture critical semantics effectively. Sparse-SDS approximates Gaussian statistics for efficient saliency estimation with minimal performance loss. In general, SDS improves progressive understanding in important regions, significantly improving segmentation in complex scenarios.

Table 3: Comparison results (mIoU, mAcc %) on the Vaihingen and Potsdam datasets. Bold indicates the best, red denotes the second-best.

| Dataset | Strategy | | | | | |
|---|---|---|---|---|---|---|
| | Random | Linear | Euclidean | Standard Gaussian | Full-SDS | Sparse-SDS |
| Vaihingen | 55.23, 74.54 | 66.05, 78.15 | 68.96, 78.02 | 67.03, 77.94 | 71.61, 79.25 | **71.81, 79.42** |
| Potsdam | 67.44, 79.93 | 71.64, 82.84 | 74.53, 84.42 | 73.54, 83.06 | **77.22**, 84.77 | 77.14, **85.01** |

## 5 CONCLUSION

In this paper, we identify the scanning order of Mamba as a key factor affecting its performance in remote sensing semantic segmentation. Inspired by the progressive perception of the human visual system, we propose VF-Mamba, which replaces fixed sequential scanning with SDS to prioritize salient regions. We use global WD as a saliency metric, based on the observation that salient regions usually exhibit larger distributional differences. To balance accuracy and efficiency, we introduce two variants: Full-SDS precisely models differences via full pairwise WD, while Sparse-SDS approximates saliency using a global Gaussian distribution. Extensive experiments on two public datasets show that VF-Mamba achieves superior performance in complex scenes.

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
