# OpenReview forum: "VF-Mamba: See What Matters First for Remote Sensing Semantic Segmentation"
_ICLR.cc/2026/Conference — ICLR 2026 Conference Withdrawn Submission_

### Official Review · Reviewer_n99u · 2025-10-27

**Soundness:** 3
**Presentation:** 3
**Contribution:** 3
**Rating:** 4
**Confidence:** 4

**Summary:**

In summary, the paper addresses an important limitation of current Mamba-based segmentation models and presents an intuitive, human-vision-inspired approach to improve scan order adaptivity. The results are promising, especially for fine-grained and rare-class segmentation. Nevertheless, the incremental nature of the architectural change, the lack of broader performance analysis, and the limited ablation coverage prevent the work from fully realizing its potential impact. Expanding the experimental scope, providing deeper discussion of comparative results, and demonstrating the generalizability of SDS would significantly strengthen the paper.

**Strengths:**

The paper presents a well-motivated modification to the Mamba scanning mechanism for remote sensing semantic segmentation. The key contribution is the introduction of a saliency-driven scanning (SDS) strategy, where the scan order is determined by patch-level saliency scores based on distributional distinctiveness. Two implementations are proposed: Full-SDS, which uses pairwise Wasserstein Distance for precise computation, and Sparse-SDS, which adopts a Gaussian-based approximation to improve efficiency. This design is novel within the context of state space models for vision, and the proposed trade-off between accuracy and computational cost is appealing. The experimental results on ISPRS Potsdam and Vaihingen datasets are strong overall, showing consistent gains over several competitive baselines.

**Weaknesses:**

1. Limited architectural novelty — The core modification is essentially the replacement of the fixed scanning order with a saliency-based ordering computed using two different distributional distance measures. While this idea is reasonable, the rest of the Mamba design remains largely unchanged, meaning that the architectural innovation is incremental rather than fundamental. A deeper integration of saliency-driven processing into the state space update mechanism could further strengthen the novelty.
2. Performance competitiveness — Although the proposed method achieves notable improvements in several metrics, DeeplabV3+ achieves comparable or even superior results for certain categories and evaluation measures. This observation suggests that the benefits of SDS may be task- or category-dependent. A deeper analysis and discussion are needed to clarify when SDS provides substantial advantages and when it struggles to outperform well-established baselines.
3. Ablation study scope — The current ablation studies primarily show that adding SDS improves performance, but they do not clearly separate the gains due to SDS itself from those already provided by the underlying Mamba architecture. More controlled experiments are necessary, such as:
- Swapping SDS into non-Mamba backbones (e.g., Transformer- or CNN-based architectures) to verify its generality.
- Systematically comparing alternative distance metrics beyond Wasserstein Distance and Gaussian approximation to evaluate whether the observed improvements are metric-specific.
- Conducting efficiency–accuracy trade-off analysis to quantify the computational overhead introduced by SDS, particularly for Full-SDS.

**Questions:**

See the weaknesses

---

### Official Review · Reviewer_jEnJ · 2025-10-30

**Soundness:** 3
**Presentation:** 3
**Contribution:** 2
**Rating:** 4
**Confidence:** 5

**Summary:**

1.	This paper presents VF-Mamba, a novel semantic segmentation model for remote sensing imagery that introduces a saliency-driven scanning (SDS) mechanism into the Mamba state space model framework.
2.	The core idea is to replace the fixed, linear scanning order of standard Mamba with a dynamic, content-aware order that prioritizes salient image patches, mimicking human visual perception.
3.	The authors propose two variants of SDS—Full-SDS and Sparse-SDS—to balance precision and computational cost. The method is evaluated on two standard ISPRS datasets (Potsdam and Vaihingen), demonstrating competitive or superior performance compared to several state-of-the-art models, particularly on rare and challenging object categories.

**Strengths:**

1.  The paper's central contribution—adapting Mamba's scanning order based on visual saliency—is innovative and well-grounded. The analogy to the human visual system is compelling and directly addresses a known limitation (fixed scanning order) of existing Mamba-based vision models. This is a meaningful step towards more spatially adaptive and efficient sequence modeling for images.
2.  The proposed SDS mechanism is elegantly designed. Using the Wasserstein Distance (WD) to measure distributional distinctiveness is a sound choice for remote sensing data, as it is more robust to complex backgrounds and geometric structures than simpler metrics like Euclidean distance. The introduction of two strategies (Full-SDS for accuracy, Sparse-SDS for efficiency) shows thoughtful consideration of practical deployment scenarios.
3. The paper provides comprehensive experimental results. The method achieves state-of-the-art or highly competitive results on both benchmark datasets, with a notable improvement on challenging categories. The consistent outperformance of other Mamba variants (VMamba, DefMamba, UNetMamba) strongly validates the effectiveness of the proposed adaptive scanning.
4.The visual results effectively demonstrate the model's strength in preserving fine details, geometric regularity, and boundary integrity, especially for small objects like vehicles.
5.The ablation study in Table 3 is excellent. It convincingly isolates the contribution of the SDS module by comparing against random, linear, and Euclidean-based scanning, and validates the design choice of using a learned Gaussian reference in Sparse-SDS.

**Weaknesses:**

1.  While the paper correctly states that Full-SDS has higher complexity (O(L²)) than Sparse-SDS (O(L)), a more detailed analysis is needed. Reporting actual training/inference times, FLOPs, or memory usage for both variants compared to a baseline (e.g., VMamba) would provide a much clearer picture of the efficiency trade-off. This is crucial for readers to choose between Full-SDS and Sparse-SDS in practice.
2.  The link between distributional distinctiveness and semantic saliency is asserted but could be more deeply explored. While the results support the claim, a brief analysis or visualization (e.g., showing the top salient patches selected by SDS) could provide more intuitive evidence that the mechanism is indeed prioritizing semantically important regions like rare objects, rather than just statistically anomalous background patches.
3.  The description of how the saliency-sorted sequence is fed into the S6 block and how the original order is "restored based on the indices" is somewhat brief. A more detailed explanation or a reference to how this is compatible with the causal nature of the Mamba SSM would be helpful.
4.  The discussion could be strengthened by explicitly stating the limitations of the proposed approach. For instance, how does the performance of Sparse-SDS degrade with highly non-Gaussian data? Is there a risk of the model becoming over-specialized on the specific statistical properties of the training data?
5.While comparison to other remote sensing Mamba models is good, a brief comparison with a recent adaptive/computationally efficient Transformer (e.g., a specific linear attention variant) could better situate the contribution within the broader field beyond Mamba-family models.

**Questions:**

1.  Could you provide a quantitative analysis (e.g., wall-clock time, FLOPs) of the computational overhead introduced by Full-SDS and Sparse-SDS compared to the base scanning mechanism?
2.  The restoration of the original order after the S6 block is a key step. Could you elaborate on how the hidden states from the scrambled sequence are mapped back to the original 2D spatial layout?
3.  Have you observed any failure cases or scenarios where the SDS mechanism does not lead to improvement, or even harms performance?

---

### Official Review · Reviewer_6DhH · 2025-10-30

**Soundness:** 3
**Presentation:** 3
**Contribution:** 2
**Rating:** 4
**Confidence:** 4

**Summary:**

The paper proposes VF-Mamba, a saliency-driven state-space model for remote sensing semantic segmentation. Its core innovation lies in integrating Saliency-Driven Scanning (SDS) into the Mamba framework, enabling progressive semantic understanding by prioritizing the scanning of information-rich image regions. The authors design two saliency scanning strategies: Full-SDS, which uses pairwise Wasserstein distances to precisely capture distributional differences between image patches, and Sparse-SDS, which employs a Gaussian reference for efficient approximation. Experiments demonstrate that this method outperforms existing approaches in preserving details of rare classes, improving IoU, and enhancing segmentation performance in complex scenes.

**Strengths:**

Strengths:
1.The paper draws inspiration from the human visual system, progressively building global semantic representations. At the same time, it integrates this mechanism into a state-space model, making the method broadly applicable.
2.The proposed method has been thoroughly validated on two public datasets, achieving strong performance.

**Weaknesses:**

Weaknesses：
1.The dataset validation is insufficient, as the paper only evaluates on two public datasets (Vaihingen and Potsdam), while there are other well-known datasets in the field, such as LoveDA.
2.The comparative experiments in 2025 only include MSEONet; more 2025 methods need to be evaluated to demonstrate the effectiveness of the experiments. Additionally, two other Mamba variants (PyramidMamba, PPMamba, SGCMamba) should be included to better showcase the effectiveness of the proposed method.
3.The core motivation of the method is relatively weak, as many existing papers have already adopted similar ideas as their central concept.

**Questions:**

1.The core motivation of VF-Mamba is saliency-driven scanning to prioritize information-rich regions; however, several existing papers in remote sensing semantic segmentation have already proposed similar saliency- or attention-driven mechanisms. The authors do not provide a theoretical analysis in the paper to demonstrate whether the SDS method in VF-Mamba offers unique advantages or substantial improvements over existing saliency or attention mechanisms.
2.The paper only evaluates the method on the Vaihingen and Potsdam datasets, whereas there are multiple other well-known datasets in remote sensing semantic segmentation, such as LoveDA. Moreover, the comparative experiments include only MSEONet and some Mamba variants (PyramidMamba, PPMamba, SGCMamba are not included). How do the authors demonstrate that the experimental results are generalizable and that the method is effective? Have they considered extending the evaluation and including more comparative methods to enhance credibility?

---

### Official Review · Reviewer_9iMq · 2025-10-31

**Soundness:** 3
**Presentation:** 2
**Contribution:** 2
**Rating:** 4
**Confidence:** 4

**Summary:**

The paper introduces VF-Mamba, a new method for semantic segmentation in remote sensing images, aimed at improving the interpretation of complex scenes. It is inspired by the human visual system, which prioritizes salient regions when interpreting scenes. The authors propose Saliency-Driven Scanning (SDS), which ranks image patches based on their distributional differences, focusing on the most informative regions first. This approach improves segmentation by efficiently capturing fine-grained details, especially in rare or complex categories.

Contributions:

1.VF-Mamba: A saliency-guided model that adapts the scanning order of patches based on their saliency, enhancing segmentation accuracy and efficiency.

2.Saliency-Driven Scanning (SDS): Two novel strategies, Full-SDS and Sparse-SDS, are introduced to compute saliency scores using Wasserstein distance (Full-SDS) and a Gaussian reference (Sparse-SDS), offering a trade-off between precision and computational efficiency.

3.State-of-the-art Performance: Extensive experiments on benchmark datasets (ISPRS Potsdam and Vaihingen) show VF-Mamba outperforms existing methods, with improvements in accuracy and segmentation of small or rare objects.

**Strengths:**

1. Novelty and Motivation:
    The idea of incorporating saliency-guided scanning into the Mamba framework is highly innovative and well-motivated. Drawing inspiration from the human visual system to prioritize salient regions for progressive scene understanding is a compelling contribution, especially in the context of state space models for vision.

2. Significant Improvement on Rare Categories:
    The method demonstrates a clear and notable improvement in the segmentation of rare and challenging categories (e.g., vehicles), with a reported 3.28% IoU gain. This addresses a critical challenge in remote sensing segmentation and is a strong practical contribution.

**Weaknesses:**

1. The paper lacks systematic overhead reporting for training/inference (time, FLOPs, parameter amounts, etc.)
2. Figure 2 (framework diagram) and Figure 3 (SDS schematic diagram) in the paper are too simple and fail to clearly show the key workflow of the algorithm.
3. The experiments of the paper are limited to the two data sets of ISPRS Potsdam and Vaihingen, and limited progress has been made in many classes on these two data sets.
4. The paper conducts a multi-category semantic segmentation task. What is the motivation for using pixel-wise binary cross-entropy loss?
5. The manuscript uses S6 inconsistently (“S6 module” vs. “S6 block”). Please standardize the terminology (use only one).
6. The Method section does not clearly describe how raw data are preprocessed/encoded into the model’s effective inputs, nor how the intermediate outputs of Full-SDS / Sparse-SDS are constructed, interfaced with downstream modules, and ultimately aggregated into the final output.

**Questions:**

As described by the weaknesses:

1. The paper lacks systematic overhead reporting for training/inference (time, FLOPs, parameter amounts, etc.)
2. Figure 2 (framework diagram) and Figure 3 (SDS schematic diagram) in the paper are too simple and fail to clearly show the key workflow of the algorithm.
3. The experiments of the paper are limited to the two data sets of ISPRS Potsdam and Vaihingen, and limited progress has been made in many classes on these two data sets.
4. The paper conducts a multi-category semantic segmentation task. What is the motivation for using pixel-wise binary cross-entropy loss?
5. The manuscript uses S6 inconsistently (“S6 module” vs. “S6 block”). Please standardize the terminology (use only one).
6. The Method section does not clearly describe how raw data are preprocessed/encoded into the model’s effective inputs, nor how the intermediate outputs of Full-SDS / Sparse-SDS are constructed, interfaced with downstream modules, and ultimately aggregated into the final output.

---

### Note · Authors · 2025-11-12

I have read and agree with the venue's withdrawal policy on behalf of myself and my co-authors.